Subject Areas:
ecology/environmental science/evolution

Keywords:
Alpine ungulates, Bergmann's rule, biometric monitoring, climate change, metabolic rate, organism shrinking

Author for correspondence:
Ulf Büntgen
e-mail: ulf.buentgen@geog.cam.ac.uk

# Stable body size of Alpine ungulates

Ulf Büntgen[1,2,3,4], Hannes Jenny[5], J. Diego Galván[2], Alma Piermattei[1], Paul J. Krusic[1,6] and Kurt Bollmann[2]

[1]Department of Geography, University of Cambridge, Downing Place CB2 3EN, UK
[2]Swiss Federal Research Institute WSL, Zürcherstr. 111, 8903 Birmensdorf, Switzerland
[3]Global Change Research Centre (CzechGlobe), 603 00 Brno, Czech Republic
[4]Department of Geography, Faculty of Science, Masaryk University, 613 00 Brno, Czech Republic
[5]Department of Wildlife and Fishery Service Grisons, Canton of Grisons, Loëstrasse 14, 7001 Chur, Switzerland
[6]Department of Physical Geography, Stockholm University, SE-10691 Stockholm, Sweden

 UB, 0000-0002-3821-0818; KB, 0000-0002-4690-7121

In many species, decreasing body size has been associated with increasing temperatures. Although climate-induced phenotypic shifts, and evolutionary impacts, can affect the structure and functioning of marine and terrestrial ecosystems through biological and metabolic rules, evidence for shrinking body size is often challenged by (i) relatively short intervals of observation, (ii) a limited number of individuals, and (iii) confinement to small and isolated populations. To overcome these issues and provide important multi-species, long-term information for conservation managers and scientists, we compiled and analysed 222 961 measurements of eviscerated body weight, 170 729 measurements of hind foot length and 145 980 measurements of lower jaw length, in the four most abundant Alpine ungulate species: ibex (*Capra ibex*), chamois (*Rupicapra rupicapra*), red deer (*Cervus elaphus*) and roe deer (*Capreolus capreolus*). Regardless of age, sex and phylogeny, the body mass and size of these sympatric animals, from the eastern Swiss Alps, remained stable between 1991 and 2013. Neither global warming nor local hunting influenced the fitness of the wild ungulates studied at a detectable level. However, we cannot rule out possible counteracting effects of enhanced nutritional resources associated with longer and warmer growing seasons, as well as the animals' ability to migrate along extensive elevational gradients in the highly diversified alpine landscape of this study.

## 1. Introduction

In line with Bergmann's rule [1] and Darwin's observation [2], decreasing body size is recognized as an important evolutionary response to increasing temperatures [3–5]. It has been reported

that shrinking body mass and size in many taxa can have an effect on the structure and functioning of marine and terrestrial ecosystems [6–11]. Understanding climate-induced changes in the weight and size of species is, therefore, an important task for scientists and conservation managers around the world. Despite strong evidence for declining avian body size in response to global warming [12–15], records of body size are restricted to a few studies that rely on either a limited number of individuals, a single species, small and isolated populations or a combination thereof [16–20]. The expected negative impacts of increasing temperatures, via heat stress and a reduction in the quality and quantity of resources, on the metabolic rate and phenotypic plasticity of animals are not well understood [21–24].

Declining morphological trait sizes have also been associated with selective hunting pressure [25–28]; the directed removal of animals with specific traits. Yet again, the lack of sufficiently long and well-replicated datasets complicates any systematic assessment of species' evolutionary responses to intensive trophy harvesting [29,30]. Hence, it remains debatable whether a properly managed hunting system can be used as a conservation tool for maintaining sustainability [31,32]. Unravelling the genetic and environmental mechanisms strong enough to influence the morphology and/or phenology in animal populations describes another important challenge [33].

Here, we analyse a large collection of eviscerated body weight (EBW), hind foot length (HFL) and lower jaw length (LJL) measurements of the four most abundant ungulate species in the European Alps. Our dataset includes precise information from each animal of the two Bovid species ibex (*Capra ibex*) and chamois (*Rupicapra rupicapra*), as well as the two Cervid species red deer (*Cervus elaphus*) and roe deer (*Capreolus capreolus*). All animals were harvested between 1991 and 2013 in the Swiss Canton of Grisons (GR). Shaping life-history traits, and thereby influencing animal fitness, skeletal measures can be used as indicators of juvenile growth and phenotypic expression later in life [34–36]; the result of a complex interplay between an organism's genes and its environment. Large and heavy individuals are expected to have higher lifetime reproductive success because big males should gain access to more females [37], and superior females should wean more offspring [38,39]. Using different regression functions and discrete time-series analyses, as well as a more conservative split-period approach, and accounting for the possible effects of age, sex, phylogeny, biometric measure and harvest elevation, we test if the body weight and size of GR's wild ungulates has changed since 1991 in response to increasing temperatures and persisting hunting pressure. Moreover, we assess if males of the most dimorphic species were affected by warming-induced changes in their nutritional resources more strongly than were their female counterparts.

## 2. Methods

GR's hunting inventory includes 222 961 EBW measurements, and two skeletal measurements, i.e. 170 729 HFL and 145 980 LJL of the sympatric ibex, chamois, red deer and roe deer, harvested from 1991 to 2013 [40] (see electronic supplementary material for details). All of these measurements were performed by official wildlife wardens and gamekeepers following strict protocols. Despite some degree of uncertainty [41], EBW is sensitive to annual changes in the animals' environment [18], whereas both skeletal measures—taken without removing the animals' skin—are useful to characterize the growth of ungulates as a combined result of genetic and ecological factors [42]. Each of the analysed 226 335 ungulates lived within approximately 7000 km$^2$ of what is considered a heterogeneous landscape comprising meadows, pastures, different forest types, alpine tundra and cliffs, as well as areas of permanent snow and ice, between 260 and 4059 m.a.s.l. (see [40] for intra- and interannual changes in harvest elevation). Permitted to the residents of GR, the hunting of free-ranging chamois, red deer and roe deer is restricted to only two to three weeks in September; and ibex to three weeks in October. No trends are found in the weight, size and age of the harvested animals within and between the species-specific autumnal hunting seasons. Moreover, trophy selection and supplementary feeding of GR's wild ungulate populations is prohibited [29].

In addition to the assessment of year-to-year changes in the number of GR's harvested male and female ungulates (electronic supplementary material, figure S1), our dataset reveals distinct, species- and sex-specific relationships between animal age, EBW, HFL and LJL (electronic supplementary material, figures S2–S4). Due to allometric constraints and growth priorities, there are positive associations between the animals' more plastic EBW (a composite expression of body size and animal vigour, with a particularly strong influence of size on the mass of young animals), and the generally more heritable HFL and LJL (electronic supplementary material, figures S5–S6). Likewise, there is a tendency for stronger linkages between the mass and size of younger and smaller animals compared

to older and larger animals. We also analysed the yearly mean, median and standard deviation of EBW, HFL, LJL and harvest elevation, for each species, by sex and age, between 1991 and 2013. Additionally, we tested for any interdependencies between the three biometric measures, as well as for possible elevational effects on the ungulates' body mass and size. All calculations were performed in R [43].

We computed the mean HFL and LJL for each species, by sex and harvesting age, over two independent, though equally long subperiods: 1991–2000 and 2004–2013. While this conservative approach ignores interannual variability, it provides a robust measure of longer-term changes in the data mean values. Annual horn growth increments were used to age the Bovids, whereas the condition of premolars and molars was used for age estimates of the Cervids. Acknowledging some degree of uncertainty in the age estimates of adult deer [44], we clustered our dataset into the following age classes: 1, 2, 3, 4–6, 7–9, 10–12 and 13+ years (only individuals over 15–16 months are considered as yearlings). The observed differences in body size between the split time-periods were validated by the non-parametric 'Mann–Kendall' and 'Wilcoxon' tests, as well as 'Holm–Bonferroni' correlation coefficients. To quantify Europe's unprecedented recent warming trend [45], monthly temperature means, centred over the eastern Swiss Alps (46.5° N, 10° E), were extracted from the latest generation of Berkeley's gridded earth surface temperature anomaly field (after [46]).

We compared monthly resolved values of the normalized differenced vegetation index (NDVI), averaged from 1 km resolution grid boxes over GR's study area [40], against the species-specific year-to-year and longer-term trends in EBW, HFL and LJL of the harvested animals. We also performed separate multiple regression analyses for each species and sex, in which EBW, HFL and LJL were used as dependent variables, and the animals' age, harvesting elevation, harvesting year, monthly mean temperature and monthly NDVI were used as independent variables. The independent variables were standardized to have a mean of zero and a standard deviation of one, and the dependent variables were log-transformed. We applied a forward/backward model selection based on the lowest Bayes information criterion (BIC). All analyses were performed in the R package 'Rcmdr' [47].

# 3. Results

Regional mean annual surface temperature anomalies increased from 0.82°C in 1991–2000 to 0.98°C in 2004–2013. During the same period of time, the autumnal hunting intensity slightly decreased from 108 856 harvested animals in 2004 to 89 238 in 2013. Year-to-year changes in EBW, HFL and LJL of each species and sex indicate stable body mass and size of GR's four most abundant wild ungulate species since 1991 (figure 1). The generally plastic and heritable measures jointly exhibit weak interannual variability, and there is no statistical evidence for significant long-term trends in any of our data. This finding is supported by our split-period analyses, independently obtained from each age class (table 1 and figure 2). When calculated over two equally long early and late split periods, we find ungulate body size in GR remained stable irrespective of age, sex, phylogeny and the biometric measure used. None of the four species at any age showed a detectable reduction in EBW, HFL and LJL (Mann–Kendall $p > 0.05$ and $t$-test $p = 1$).

Our analysis further reveals that ibex, chamois and red deer males are overall larger than same-aged females (figure 2), whereas roe deer body sizes of both sexes are similar. The LJL of male ibex and red deer is larger than that of females of the same age, but tends to be nearly identical between male and female chamois and roe deer. Both morphological traits reveal comparable growth patterns, with the smallest juvenile increase in roe deer and chamois, and the most distinct and prolonged growth trend in ibex and red deer. The sexual dimorphism ratio of HFL in adults is largest for ibex (1.15), followed by chamois and red deer (both 1.06). Young males of all dimorphic species tend to exhibit a faster increase in HFL compared to their female counterparts (figure 2 and table 1). The multiple regression analysis confirms that age has the strongest effect on EBW, HFL and LJL in all species and sexes (electronic supplementary material, tables S1–S4), followed by harvest elevation (electronic supplementary material, figures S7–S10). The EBW and LJL of both ibex and chamois males and females are positively related to their harvest elevation, whereas this overall positive relationship is less pronounced for the Bovids' HFL. It appears that the body mass and size of red deer tend to be less dependent on harvest elevation. Statistically meaningful and temporally stable relationships are not found between any of the animals' post-mortem measurements of EBW, HFL and LJL and high-resolution climate and NDVI variables.

R. Soc. Open Sci. 7: 200196

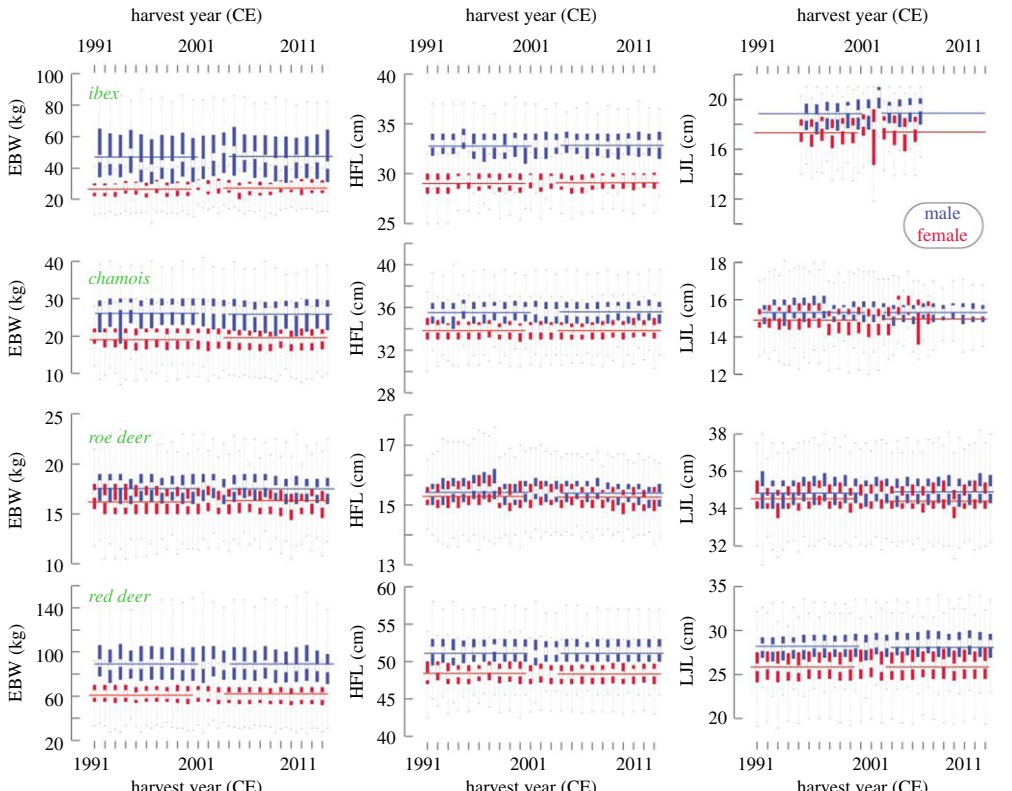

**Figure 1.** Year-to-year (vertical bars from 1991–2013) and long-term (horizontal lines spanning the two subperiods 1991–2000 and 2004–2013) changes in eviscerated body weight (EBW), hind foot length (HFL) and lower jaw length (LJL) of Grisons' wild ungulates. Box plots show the distribution range of all data per year and age, with horizontal white lines representing the median, and the upper/lower box limits referring to the 75th/25th percentile. Whiskers extend from the two ends of the box until they reach the sample maximum and minimum. Outliers are not shown to facilitate the reading. Horizontal lines show the species-specific, subperiod grand average of all harvested male and female ungulates. The y-axes have slightly different scales to facilitate visual comparison. Performing the same analysis for each age class (4 × 2 × 7) resulted in 56 cases that showed no differences between the two subperiods (figure 2 and table 1 for details).

# 4. Discussion and conclusion

Unlike previous work [3–14,16–20], this study finds body mass and size of wild ungulates was stable during the past decades, when temperatures increased [45]. During the same period of time, GR's persistent management of natural resources resulted in approximately constant population sizes of each species studied [29,40].

Explanations for our inability to detect any significant, long-term changes in both the more plastic as well as more heritable traits in GR's harvested wild ungulates are manifold. Despite the fact that our study was conducted during Europe's warmest period of the past two millennia [48], the observational interval might be too short, and/or with too little warming in between [49]. Another explanation for the apparent stable body size of Alpine ungulates could be the particularly slow evolutionary and phenotypic responses in species with long generation times [50], as opposed to faster environmental effects [17,33,51]. Moreover, it is possible that warming-induced changes in the size and weight of the species have happened previously and the animals have reached their adaptive trough. Due to relatively low reproduction rates and harvesting ages, and thus the species' overall high turnover rate, we expect there will be detectable changes in body weight and size within a few decades. Since we analysed harvest data from animals that inhabit lower, intermediate or higher altitude areas, temporal changes in the elevational movement of ungulates may mask some of the expected responses [40]. In addition, positive environmental factors, such as more and better nutrition during longer and warmer vegetation periods, may counterbalance some of the negative effects of recent warming and selective hunting on the body size of Alpine ungulates [52]. Since the quality and quantity of forage is jointly determined by climate variation and grassland management, including

**Table 1.** Comparison of long-term mean and variance changes in eviscerated body weight (EBW, in kg), hind foot length (HFL, in cm) and lower jaw length (LJL, in cm) of Grisons' wild ungulates between the two subperiods 1991–2000 and 2004–2013. While the first value always refers to the temporal average, and the second values are the corresponding standard deviations. None of the 192 pairings reveals any significant difference between the early and late interval.

| species | variable | sex | mean (1991–2000) ±s.d. (all) | mean (2004–2013) ±s.d. (all) | mean (1991–2000) ±s.d. (1 yr age class) | mean (2004–2013) ±s.d. (1 yr age class) | mean (1991–2000) ±s.d. (2–4 yrs age class) | mean (2004–2013) ±s.d. (2–4 yrs age class) | mean (1991–2000) ±s.d. (>8 yrs age class) | mean (2004–2013) ±s.d. (>8 yrs age class) |
|---|---|---|---|---|---|---|---|---|---|---|
| ibex | EBW | female | 26.49 ± 6.06 | 27.17 ± 6.09 | 16.89 ± 2.67 | 17.26 ± 2.56 | 25.29 ± 3.96 | 25.96 ± 3.99 | 30.68 ± 4.18 | 31.18 ± 4.16 |
| | EBW | male | 46.54 ± 17.37 | 47.24 ± 17.1 | 17.56 ± 3.52 | 18.93 ± 3.57 | 36.28 ± 8.21 | 35.83 ± 8.27 | 68.20 ± 8.64 | 67.41 ± 8.45 |
| | HFL | female | 29.04 ± 1.38 | 29.12 ± 1.43 | 27.25 ± 1.17 | 27.39 ± 1.09 | 29.05 ± 1.15 | 29.11 ± 1.25 | 29.65 ± 1.03 | 29.71 ± 1.03 |
| | HFL | male | 32.68 ± 2.04 | 32.74 ± 1.93 | 28.11 ± 1.45 | 28.36 ± 1.26 | 32.26 ± 1.65 | 32.16 ± 1.51 | 33.94 ± 1.17 | 34.04 ± 1.15 |
| | LJL | female | 29.04 ± 1.47 | 29.12 ± 1.63 | 15.11 ± 0.81 | 14.96 ± 0.72 | 17.05 ± 0.95 | 17.34 ± 1.06 | 18.6 ± 0.75 | 18.61 ± 0.71 |
| | LJL | male | 32.68 ± 1.62 | 32.74 ± 1.45 | 15.51 ± 0.98 | 16.95 ± 1.19 | 18.11 ± 1.18 | 18.28 ± 1.08 | 20.03 ± 0.81 | 20.05 ± 0.78 |
| chamois | EBW | female | 19.44 ± 3.66 | 18.91 ± 3.57 | 15.02 ± 2.29 | 15.28 ± 2.18 | 19.98 ± 2.67 | 19.90 ± 2.47 | 21.23 ± 2.73 | 21.09 ± 2.56 |
| | EBW | male | 25.58 ± 5.99 | 25.05 ± 5.94 | 15.34 ± 2.35 | 15.36 ± 2.41 | 26.41 ± 3.81 | 25.48 ± 3.87 | 28.83 ± 3.27 | 28.56 ± 3.23 |
| | HFL | female | 33.79 ± 1.43 | 33.82 ± 1.35 | 32.66 ± 1.42 | 32.93 ± 1.34 | 34.09 ± 1.23 | 34.16 ± 1.08 | 34.19 ± 1.17 | 34.35 ± 1.09 |
| | HFL | male | 35.43 ± 1.66 | 35.45 ± 1.63 | 33.46 ± 1.49 | 33.52 ± 1.55 | 35.80 ± 1.26 | 35.75 ± 1.22 | 35.97 ± 1.43 | 36.73 ± 1.21 |
| | LJL | female | 14.99 ± 0.95 | 15.02 ± 1.04 | 13.78 ± 0.64 | 14.02 ± 0.79 | 15.04 ± 0.65 | 15.33 ± 0.57 | 15.63 ± 0.61 | 15.69 ± 0.61 |
| | LJL | male | 15.34 ± 0.97 | 15.19 ± 0.83 | 13.87 ± 0.67 | 13.8 ± 0.56 | 15.51 ± 0.66 | 15.31 ± 0.55 | 15.86 ± 0.66 | 15.66 ± 0.49 |
| roe deer | EBW | female | 16.37 ± 2.07 | 16.03 ± 2.12 | 15.56 ± 1.76 | 15.46 ± 1.63 | 16.9 ± 1.8 | 16.94 ± 1.63 | 17.08 ± 1.95 | 16.97 ± 1.95 |
| | EBW | male | 17.48 ± 2.05 | 17.43 ± 2.04 | 15.98 ± 1.72 | 16.14 ± 1.68 | 18.02 ± 1.71 | 18.11 ± 1.63 | 17.97 ± 1.84 | 17.99 ± 2.03 |
| | HFL | female | 34.55 ± 1.29 | 34.5 ± 1.3 | 34.35 ± 1.17 | 34.40 ± 1.04 | 34.8 ± 1.07 | 34.87 ± 1.02 | 34.89 ± 1.02 | 35.06 ± 0.96 |
| | HFL | male | 34.94 ± 1.28 | 34.94 ± 1.16 | 34.73 ± 1.21 | 34.73 ± 1.09 | 35.07 ± 1.2 | 35.13 ± 1.05 | 35.4 ± 0.86 | 35.04 ± 1.16 |
| | LJL | female | 15.34 ± 0.68 | 15.24 ± 0.63 | 15.02 ± 0.57 | 14.99 ± 0.48 | 15.53 ± 0.57 | 15.46 ± 0.48 | 15.74 ± 0.58 | 15.59 ± 0.54 |
| | LJL | male | 15.48 ± 0.65 | 15.39 ± 0.56 | 15.05 ± 0.6 | 15.01 ± 0.49 | 15.61 ± 0.56 | 15.54 ± 0.48 | 15.76 ± 0.67 | 15.62 ± 0.47 |
| red deer | EBW | female | 61.58 ± 11.17 | 60.16 ± 11.55 | 55.98 ± 6.71 | 54.69 ± 6.46 | 65.71 ± 7.39 | 64.84 ± 7.21 | 73.44 ± 8.29 | 73.40 ± 8.41 |
| | EBW | male | 90.03 ± 24.87 | 89.04 ± 25.31 | 63.17 ± 8.88 | 61.63 ± 8.95 | 91.37 ± 14.15 | 88.78 ± 13.19 | 133.63 ± 19.58 | 135.52 ± 20.91 |
| | HFL | female | 48.35 ± 2.38 | 48.22 ± 2.27 | 47.80 ± 20.6 | 47.77 ± 1.83 | 48.93 ± 1.95 | 48.86 ± 1.83 | 49.75 ± 1.88 | 49.57 ± 1.79 |
| | HFL | male | 51.1 ± 2.77 | 51.12 ± 2.62 | 49.3 ± 2.23 | 49.22 ± 2.34 | 51.62 ± 2.15 | 51.61 ± 1.94 | 52.8 ± 1.98 | 52.74 ± 2.04 |
| | LJL | female | 25.81 ± 2.29 | 25.96 ± 2.36 | 24.43 ± 1.15 | 24.77 ± 1.23 | 26.79 ± 1.22 | 27.06 ± 1.26 | 28.44 ± 1.15 | 28.75 ± 1.14 |
| | LJL | male | 27.85 ± 2.49 | 28.2 ± 2.51 | 25.28 ± 1.44 | 25.53 ± 1.45 | 28.35 ± 1.32 | 28.66 ± 1.35 | 30.76 ± 1.46 | 31.07 ± 1.46 |

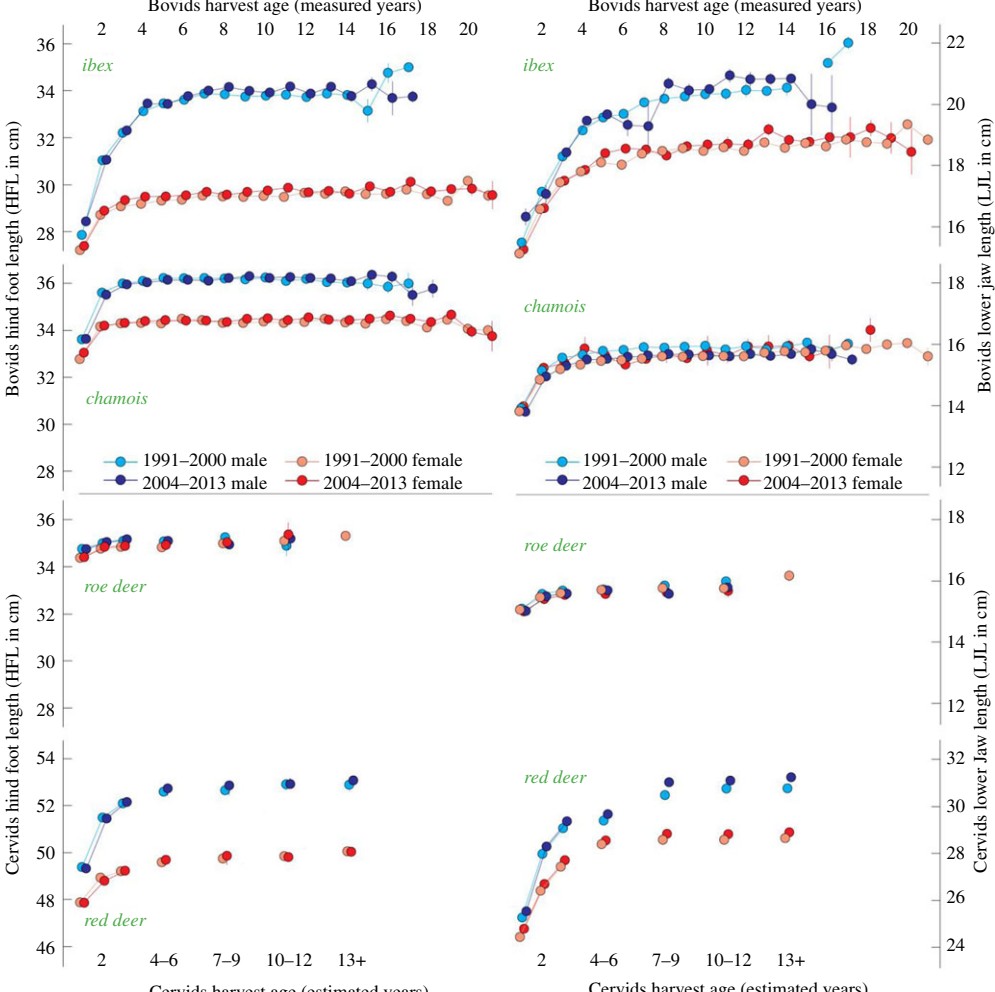

**Figure 2.** Long-term changes in hind foot length (HFL) and lower jaw length (LJL) of Grisons' wild ungulates between the two subperiods 1991–2000 and 2004–2013. Vertical lines indicate standard errors. The y-axes have slightly different scales to facilitate visual comparison (see electronic supplementary material, for details).

the use of fertilizers at lower elevations [53], their covariance may blur a direct negative relationship, that between increasing ambient temperature and shrinking body size due to thermoregulatory efficiency. Although our findings are independent of sex and species (selection pressure should be largest for ibex and red deer males but lowest for roe deer females), some additional bias may originate from the general tendency of targeting larger animals within a given age class [29], and warming-induced elevational range shifts [40].

Phenotypic plasticity of behavioural, physiological and morphological traits enables organisms to cope with environmental changes based on the natural history a species has experienced since the Pleistocene [54]. Such changes are considered to impose selection pressure on traits important for fitness [33,55], such as body size. However, here we have not found any evidence for significant long-term changes in the annual mean values of EBW, HFL and LJL in any of the four species. This suggests that the responsible mechanism of these traits has remained unchanged during the study period. This conclusion corresponds with those of *Ovis canadensis* [56] and *Cervus elaphus* [57], which claim selection has not depleted additive genetic variance in morphological traits. Skeletal structures particularly depend on the allocation of resources for growth during the juvenile life stage [58]. At the adult stage, skeletal size remains stable in most endotherms [52] and agrees with data presented here.

Exhibiting similar results, the longer HFL that range from approximately 28 to 54 cm and the shorter LJL measurements that range from approximately 16 to 32 cm (both recorded at a resolution of 1.0 mm), are indicative for the accuracy of the herein-analysed biometric parameters [59]. Since HFL and LJL are age-specific and easy to measure, they have advantages over EBW, which may exhibit more intra- and interannual variability due to environmental-induced changes in the animals' nutritional constitution

[42]. In agreement with Becciolini *et al.* [60], the HFL and LJL of GR's red deer population mainly increase during the juvenile phase. Unexpectedly and in contrast to Pettorelli *et al.* [61,62], our study does not reveal any direct effects of NDVI on interannual to decadal changes in the weight and size of GR's most abundant ungulate species since 1991. Moreover, our findings of juvenile chamois also contrast the results of Mason *et al.* [18], who reported shrinking juvenile body mass in three neighbouring chamois populations in the Italian Alps due to indirect environmental effects on resource productivity and phenology.

The magnitude of environmental drivers, such as temperature variability and food availability, are not the same for all four species in this study due to species-specific ecological requirements [40]. Heat dissipation by the smaller roe deer, for instance, should be better than by ibex or red deer. On the other hand, a better surface-to-size ratio is beneficial for sustaining cold winter seasons that last longer than the snow-free summers in Alpine environments [63]. Varying in space and time, these opposing drivers may counterbalance the directional selection of morphometric growth traits in the studied species.

Last but not least, we have to emphasize that the complex topography of the eastern Swiss Alps allows all four ungulate species to migrate along extensive elevational gradients, and thus between different climate and vegetation zones at small spatial scales [40,54]. High intra- and interannual mobility can help to mitigate a possible climate-induced selection pressure.

Data accessibility. All data are provided in the electronic supplementary material (AlpineUngulates2013.xlsx).

Authors' contributions. U.B. conceived and designed the research, analysed the data and wrote the article. H.J. compiled the data and provided critical input. J.D.G. and A.P. analysed the data and contributed to the discussion. P.J.K. and K.B. co-wrote the article. All authors contributed to discussion and approved submission.

Competing interests. We declare we have no competing interests.

Funding. This study was partly supported by the SustES project (CZ.02.1.01/0.0/0.0/16_019/0000797) of the Czech Republic.

Acknowledgements. We are very thankful to the numerous hunters, wildlife wardens and gamekeepers who contributed enthusiastically to the development of Grisons' long-term hunting inventory. Marco Festa-Bianchet and Atle Mysterud kindly commented on earlier versions of this manuscript.

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
