## [Reviewer comments · Royal Society Open Science]

Review History

RSOS-200196.R0 (Original submission)

Review form: Reviewer 1

Is the manuscript scientifically sound in its present form?

Yes

Are the interpretations and conclusions justified by the results?

Yes

Is the language acceptable?

Yes

Do you have any ethical concerns with this paper?

No

Have you any concerns about statistical analyses in this paper?

Yes

Recommendation?

Major revision is needed (please make suggestions in comments)

Comments to the Author(s)

See Appendix A.

Review form: Reviewer 2**Is the manuscript scientifically sound in its present form?**

Yes

Are the interpretations and conclusions justified by the results?

No

Is the language acceptable?

Yes

Do you have any ethical concerns with this paper?

Yes

Have you any concerns about statistical analyses in this paper?

Yes

Recommendation?

Major revision is needed (please make suggestions in comments)

Comments to the Author(s)

Review RSOS-200196

General

The impact of changes in climatological conditions on animal morphology, physiology, behaviour and life history requires the analysis of continuous, large, long-term data sets. These are particularly hard to find for large mammals. In this manuscript, the authors analyse long-term datasets for 4 artiodactyl species from a relatively small canton in the Swiss alps. The time period for the data runs from 1991 - 2013 with the data on body weight and morphology derived from animals culled in apparently well proscribed methodologies over this period. In the analysis the authors control for age, sex across the time period of the analyses. They find that there is no apparent change in weight of size of any of the species, ages or sex class across the period of investigation. This is a robust result and agrees with some other studies and disagrees with others. The authors conclude that the species studied in this context are able to move across landscapes (a mosaic of different land use and habitat types) and elevation in the light of changes in climatic conditions. This is an important and valuable study because of the size of the dataset, across 4 artiodactyl species. The manuscript is generally well written, although there are areas that require clarification and a number of editorial changes that should be made to English (see attached PDF (Appendix B)). There is one area of improvement that can be made, that is that the topography of the region apparently allows the ungulates to ameliorate the effects of temperature increases over the study period by changing elevation and most probably habitat/land use type use. The authors should present an analysis of the interaction between elevation and year on body weight and size for males and females of different ages in their 4 study species. This would allow them to assess whether there is an effect within and across elevation over time.

Specific (also see the attached PDF for edits to the manuscript)

l. 34-35. Latin names for species as well please

l. 53-54. Why is it an "important task for conservation managers"? This is a very presumptive statement. I can understand that it is scientifically interesting and may be important for sport hunters but conservation managers?

- l. 55. See also recent work on nightingales. <https://academic.oup.com/auk/advance-article-abstract/doi/10.1093/auk/ukaa012/5814682>
- l. 61 When would "Knowledge" be "complete"?
- l. 68. I am not sure how changes in weight/size with climate links to the management of biodiversity in general; maybe gene frequencies in a population/species.
- l. 70-73. A complex sentence; breakdown & redraft
- l. 99-101. References require for statements wrt the interannual variation in weight & the genetic/environmental condition on skeletal characteristics.
- L. 104-105. Is "local resident" hunting different from the hunting by wardens? I assume that the records from the "permitted" hunts are not included in the dataset.
- l. 124-125. Why were these age groupings chosen?
- l. 125. "Only individuals older than 15-16 months were considered" but you have just said you clustered "animals > 3 year.
- l. 145-148. Move this to the beginning of the Results to provide climatological background.
- l. 159-171. This is of limited importance for the manuscript. Reduce substantially (mainly the elevavtion effect should remain [see comment in General above]).
- l. 180. How do you know that the hunting system is "sustainable"?
- l. 201. Or the effects on the weight/size of the species have already happened and the species have reached it adaptive trough wrt weight & size?
- l. 210. Is the increase in fertiliser happening in this context? Wouldn't it have shown up in the NDVI?
- l. 236-238. This is probably the most important statement in the Discussion & Conclusions. This needs to be emphasised

Decision letter (RSOS-200196.R0)

Dear Dr Büntgen,

The editors assigned to your paper ("Stable body size of Alpine ungulates") have now received comments from reviewers. We would like you to revise your paper in accordance with the referee and Associate Editor suggestions which can be found below (not including confidential reports to the Editor). Please note this decision does not guarantee eventual acceptance.

Please submit a copy of your revised paper before 21-May-2020. Please note that the revision deadline will expire at 00.00am on this date. If we do not hear from you within this time then it will be assumed that the paper has been withdrawn. In exceptional circumstances, extensions may be possible if agreed with the Editorial Office in advance. We do not allow multiple rounds of revision so we urge you to make every effort to fully address all of the comments at this stage. If deemed necessary by the Editors, your manuscript will be sent back to one or more of the original reviewers for assessment. If the original reviewers are not available, we may invite new reviewers.

When submitting your revised manuscript, you must respond to the comments made by the

referees and upload a file "Response to Referees" in "Section 6 - File Upload". Please use this to document how you have responded to the comments, and the adjustments you have made. In order to expedite the processing of the revised manuscript, please be as specific as possible in your response.

- Data accessibility

If you wish to submit your supporting data or code to Dryad (<http://datadryad.org/>), or modify your current submission to dryad, please use the following link:
<http://datadryad.org/submit?journalID=RSOS&manu=RSOS-200196>

- Competing interests

- Authors' contributions

- Acknowledgements

- Funding statement

on behalf of Professor Matthew Collins (Associate Editor) and Pete Smith (Subject Editor)
openscience@royalsociety.org

Associate Editor's comments (Professor Matthew Collins):
Comments to the Author:

Both reviewers believe that the MS is a worthwhile publication, but both suggest major reviews.

The primary concern of Reviewer#1 was the statistical framework and suggests using multiple regressions instead. I would ask you to consider this suggestion. Please note that if you choose not to adopt the strategy proposed by Reviewer#1, and given that this may be raised by other readers, I think that it would also be useful to include this justification in your methods section as well as in your response to me.

Both Reviewer#1 and Reviewer#2 would also like more information on the spatial and topographic variation in the dataset.

I have added further editorial comments to the PDF supplied by Reviewer #2 which you should find attached. (RSOS-200196_Proof_hi ijg.pdf)

Reviewer#1

I reviewed the manuscript entitled "Stable body size of Alpine ungulates" and I found the problem raised in the manuscript both interesting and important.

In my opinion it will be a valuable contribution to Royal Society Open Science as the manuscript is well written and structured.

However I found problems with the applied statistical approach. In my opinion, the statistical framework should be changed to make results clearer and easier to follow.

Instead of running multiple tests on different subsets of data I propose to build separate models (multiple regressions) for every studied species with EBW, HFL LJL as dependent variables and the year, age, altitude, NDVI value as dependent variables.

The authors did not mention anything about spatial distribution of sampling that can induce spatial autocorrelation in model residuals. Unfortunately, many statements in the results section (L 110-112.) are not supported by any statistical testing

I do not understand the reasoning provided in this sentence. Fig S2, S3, S4. Why the difference in the age of roe deer given in 1-year intervals, when in methods and in Fig 2 you provide that you estimated age for this species in broader age classes?

Reviewer #2

General

The impact of changes in climatological conditions on animal morphology, physiology, behaviour and life history requires the analysis of continuous, large, long-term data sets. These are particularly hard to find for large mammals. In this manuscript, the authors analyse long-term datasets for 4 artiodactyl species from a relatively small canton in the Swiss alps. The time period for the data runs from 1991 – 2013 with the data on body weight and morphology derived from animals culled in apparently well proscribed methodologies over this period. In the analysis the authors control for age, sex across the time period of the analyses.

They find that there is no apparent change in weight of size of any of the species, ages or sex class across the period of investigation. This is a robust result which agrees with some other studies and disagrees with others.

The authors conclude that the species studied in this context are able to move across landscapes (a mosaic of different land use and habitat types) and elevation in the light of changes in climatic conditions.

This is an important and valuable study because of the size of the dataset, across 4 artiodactyl species.

The manuscript is generally well written, although there are areas that require clarification and a number of editorial changes that should be made to English (see attached PDF (RSOS-200196_Proof_hi ijpg.pdf)).

There is one area of improvement that can be made, that is that the topography of the region apparently allows the ungulates to ameliorate the effects of temperature increases over the study period by changing elevation and most probably habitat/land use type use. The authors should present an analysis of the interaction between elevation and year on body weight and size for males and females of different ages in their 4 study species. This would allow them to assess whether there is an effect within and across elevation over time.

Specific (also see the attached PDF for edits to the manuscript)

l. 34-35. Latin names for species as well please

l. 53-54. Why is it an "important task for conservation managers"? This is a very presumptive statement. I can understand that it is scientifically interesting and may be important for sport hunters but conservation managers?

l. 55. See also recent work on nightingales. <https://academic.oup.com/auk/advance-article-abstract/doi/10.1093/auk/ukaa012/5814682>

l. 61 When would "Knowledge" be "complete"?

l. 68. I am not sure how changes in weight/size with climate links to the management of biodiversity in general; maybe gene frequencies in a population/species.

l. 70-73. A complex sentence; breakdown & redraft

l. 99-101. References require for statements wrt the interannual variation in weight & the genetic/environmental condition on skeletal characteristics.

L. 104-105. Is "local resident" hunting different from the hunting by wardens? I assume that the records from the "permitted" hunts are not included in the dataset.

l. 124-125. Why were these age groupings chosen?

l. 125. "Only individuals older than 15-16 months were considered" but you have just said you clustered "animals > 3 year.

l. 145-148. Move this to the beginning of the Results to provide climatological background.

l. 159-171. This is of limited importance for the manuscript. Reduce substantially (mainly the elevation effect should remain [see comment in General above]).

- l. 180. How do you know that the hunting system is "sustainable"?
- l. 201. Or the effects on the weight/size of the species have already happened and the species have reached it adaptive trough wrt weight & size?
- l. 210. Is the increase in fertiliser happening in this context? Wouldn't it have shown up in the NDVI?
- l. 236-238. This is probably the most important statement in the Discussion & Conclusions. This needs to be emphasised

Reviewers' Comments to Author:

Reviewer: 1

Comments to the Author(s)

I reviewed the manuscript entitled "Stable body size of Alpine ungulates" and I found the problem raised in the manuscript both interesting and important.

In my opinion it will be a valuable contribution to Royal Society Open Science as the manuscript is well written and structured.

However I found problems with the applied statistical approach. In my opinion, the statistical framework should be changed to make results clearer and easier to follow.

Instead of running multiple tests on different subsets of data I propose to build separate models (multiple regressions) for every studied species with EBW, HFL LjL as dependent variables and the year, age, altitude, NDVI value as dependent variables.

The authors did not mention anything about spatial distribution of sampling that can induce spatial autocorrelation in model residuals. Unfortunately, many statements in the results section (L 110-112.) are not supported by any statistical testing

I do not understand the reasoning provided in this sentence. Fig S2, S3, S4. Why the difference in the age of roe deer given in 1-year intervals, when in methods and in Fig 2 you provide that you estimated age for this species in broader age classes?

Reviewer: 2

Comments to the Author(s)

Review RSOS-200196

General

The impact of changes in climatological conditions on animal morphology, physiology, behaviour and life history requires the analysis of continuous, large, long-term data sets. These are particularly hard to find for large mammals. In this manuscript, the authors analyse long-term datasets for 4 artiodactyl species from a relatively small canton in the Swiss alps. The time period for the data runs from 1991 - 2013 with the data on body weight and morphology derived from animals culled in apparently well proscribed methodologies over this period. In the analysis the authors control for age, sex across the time period of the analyses. They find that there is no apparent change in weight of size of any of the species, ages or sex class across the period of investigation. This is a robust result and agrees with some other studies and disagrees with others. The authors conclude that the species studied in this context are able to move across landscapes (a mosaic of different land use and habitat types) and elevation in the light of changes in climatic conditions. This is an important and valuable study because of the size of the dataset, across 4 artiodactyl species. The manuscript is generally well written, although there are areas that require clarification and a number of editorial changes that should be made to English (see attached PDF (RSOS-200196_Proof_hi ijj.pdf)). There is one area of improvement that can be

made, that is that the topography of the region apparently allows the ungulates to ameliorate the effects of temperature increases over the study period by changing elevation and most probably habitat/land use type use. The authors should present an analysis of the interaction between elevation and year on body weight and size for males and females of different ages in their 4 study species. This would allow them to assess whether there is an effect within and across elevation over time.

Specific (also see the attached PDF for edits to the manuscript)

- I. 34-35. Latin names for species as well please
- I. 53-54. Why is it an "important task for conservation managers"? This is a very presumptive statement. I can understand that it is scientifically interesting and may be important for sport hunters but conservation managers?
- I. 55. See also recent work on nightingales. <https://academic.oup.com/auk/advance-article-abstract/doi/10.1093/auk/ukaa012/5814682>
- I. 61 When would "Knowledge" be "complete"?
- I. 68. I am not sure how changes in weight/size with climate links to the management of biodiversity in general; maybe gene frequencies in a population/species.
- I. 70-73. A complex sentence; breakdown & redraft
- I. 99-101. References require for statements wrt the interannual variation in weight & the genetic/environmental condition on skeletal characteristics.
- L. 104-105. Is "local resident" hunting different from the hunting by wardens? I assume that the records from the "permitted" hunts are not included in the dataset.
- I. 124-125. Why were these age groupings chosen?
- I. 125. "Only individuals older than 15-16 months were considered" but you have just said you clustered "animals > 3 year.
- I. 145-148. Move this to the beginning of the Results to provide climatological background.
- I. 159-171. This is of limited importance for the manuscript. Reduce substantially (mainly the elevavtion effect should remain [see comment in General above]).
- I. 180. How do you know that the hunting system is "sustainable"?
- I. 201. Or the effects on the weight/size of the species have already happened and the species have reached it adaptive trough wrt weight & size?
- I. 210. Is the increase in fertiliser happening in this context? Wouldn't it have shown up in the NDVI?
- I. 236-238. This is probably the most important statement in the Discussion & Conclusions. This needs to be emphasised

Author's Response to Decision Letter for (RSOS-200196.R0)

See Appendices C & D.

Decision letter (RSOS-200196.R1)

Dear Dr Büntgen,

It is a pleasure to accept your manuscript entitled "Stable body size of Alpine ungulates" in its current form for publication in Royal Society Open Science.

on behalf of Professor Matthew Collins (Associate Editor) and Pete Smith (Subject Editor)
openscience@royalsociety.org

Associate Editor Comments to Author (Professor Matthew Collins):

Associate Editor

Comments to the Author:

Thank you for your revised version, which has fully addressed the comments raised by the referees. Please accept my apologies for the delay in assessing your revised version.

One minor typo has leaked through in what is otherwise an excellent revised version.

exhibits -> exhibits

Since HFL and LJJ are age-specific and easy to measure, they have advantages over EBW, which may exhibit more intra- and interannual variability due to environmental-induced changes in the animals' nutritional constitution (Toïgo et al., 2006).

Appendix A

I reviewed the manuscript entitled “Stable body size of Alpine ungulates” and I found the problem risen in the manuscript interesting and important. In my opinion it will be a valuable contribution to Royal Society Open Science as the manuscript is well written and structured. However I found the problem with applied statistical approach. In my opinion statistical framework should be changed to make results clearer and easier to follow. Instead of running multiple tests on different subsets of data I propose to build separate models (multiple regressions) for every studied species with EBW, HFL L/L as dependent variables and the year, age, altitude, NDVI value as independent variables. Authors did not mention anything about spatial distribution of sampling that can induce spatial autocorrelation in model residuals. Unfortunately, now many statements in result section are not supported by any statistical testing.

L 110-112. I do not understand the reasoning provided in this sentence.

Fig S2, S3, S4. Why the difference in the age roe deer is in 1-year, when in methods and in Fig 2 you provide that you estimated age for this species in broader age classes?

Appendix B**ROYAL SOCIETY
OPEN SCIENCE****Stable body size of Alpine ungulates**

Journal:	Royal Society Open Science
Manuscript ID	RSOS-200196
Article Type:	Research
Date Submitted by the Author:	07-Feb-2020
Complete List of Authors:	Büntgen, Ulf; University of Cambridge, Department of Geography; Swiss Federal Research Institute WSL, Dendrosciences; Global Change Research Centre and Masaryk University Jenny, Hannes; Department of Wildlife and Fishery Service Grison Galvan, Diego; Swiss Federal Institute for Forest Snow and Landscape Research Piermattei, Alma; University of Cambridge Krusic, Paul; University of Cambridge Bollmann, Kurt; Swiss Federal Research Institute WSL
Subject:	ecology < BIOLOGY, environmental science < BIOLOGY, evolution < BIOLOGY
Keywords:	Alpine ungulates, Bergmann's rule, biometric monitoring, climate change, metabolic rate, organism shrinking
Subject Category:	Ecology, Conservation, and Global Change Biology

Author-supplied statements

Relevant information will appear here if provided.

Ethics

Does your article include research that required ethical approval or permits?:

This article does not present research with ethical considerations

Statement (if applicable):

CUST_IF_YES_ETHICS :No data available.

Data

It is a condition of publication that data, code and materials supporting your paper are made publicly available. Does your paper present new data?:

Yes

Statement (if applicable):

All data are provided in the supplementary materials of this submission (AlpineUngulates2013.xlsx).

Conflict of interest

I/We declare we have no competing interests

Statement (if applicable):

CUST_STATE_CONFLICT :No data available.

Authors' contributions

This paper has multiple authors and our individual contributions were as below

Statement (if applicable):

UB conceived and designed the research, analysed the data and wrote the article. HJ compiled the data and provided critical input. JDG and AP analysed the data and contributed to discussion. PJK and KB co-wrote the article. All authors contributed to discussion and approved submission.

**Running Head: *Ungulate body size***

**Stable body size of Alpine ungulates**

ULF BÜNTGEN,^{1,2*} HANNES JENNY,³ J. DIEGO GALVÁN,² ALMA PIERMATTEI,¹ PAUL J.

KRUSIC,^{1,4} and KURT BOLLMANN²

*¹Department of Geography, University of Cambridge, Downing Place, CB2 3EN, UK*

*²Swiss Federal Research Institute WSL, Zürcherstr 111, 8903 Birmensdorf, Switzerland*

*³Department of Wildlife and Fishery Service Grisons, Loëstrasse 14, 7001 Chur, Switzerland*

*⁴Department of Physical Geography, Stockholm University, SE-10691, Stockholm, Sweden*

**Author for Correspondence: ulf.buentgen@geog.cam.ac.uk (Ulf Büntgen)*

Submitted as an original research *Article* to **Royal Society Open Science**, 4th February 2020

**Abstract** Decreasing body size ~~in many taxa~~ has been associated ~~recently~~ with increasing
temperatures. Although climate-induced phenotypic shifts and evolutionary impacts can affect
the structure and functioning of marine and terrestrial ecosystems through biological and
metabolic rules, evidence for shrinking body size is often challenged by relatively short
observations of a limited number of individuals in ~~sometimes~~ small and isolated populations.
To overcome ~~this conundrum~~ and provide important multi-species, long-term information for
conservation managers and scientists, we compile and analyse 222,961 measurements of
eviscerated body weight, 170,729 measurements of hind foot length, and 145,980
measurements of lower jaw length, in the four most abundant Alpine ungulate species: ibex,
chamois, red deer and roe deer. Regardless of age, sex and phylogeny, the body mass and size
of these sympatric animals from the eastern Swiss Alps remained stable between 1991 and
2013. While neither global warming nor local hunting influenced the fitness of ~~Grisons'~~ wild
ungulates at a detectable level, we cannot rule out possible counteracting effects of enhanced
nutritional resources associated with longer and warmer growing seasons in the highly
diversified alpine landscape of this study.

**Keywords** Alpine ungulates, Bergmann's rule, biometric monitoring, climate change,
metabolic rate, organism shrinking

**Introduction**

In line with Bergmann's rule (Bergmann, 1847) and Darwin's observation (Darwin, 1859),
decreasing body size is recognised as an important evolutionary response to increasing
temperatures (Sheridan & Bickford, 2011; Caruso et al., 2015; Tseng et al., 2017). Reported for
many taxa (Gardner et al., 2011; Horne et al., 2017), shrinking body mass and size may affect
the structure and functioning of marine and terrestrial ecosystems (Gibert & DeLong, 2014;

Legagneux et al., 2014; MacLean & Beissinger, 2017; Lindmark et al., 2018). Understanding
changes in the weight and size of species therefore describes an important task for conservation
managers and scientists around the world. Despite strong evidence for declining avian body
size in response to global warming (Yom-Tov, 2001; Gardner et al., 2009, 2014), records of
body size are restricted to a few studies that sometimes rely on either a limited number of
individuals, a single species, small and isolated populations, or a combination thereof (Ozgul
et al., 2009; Rughetti et al., 2012; Mason et al., 2014; van Gils et al., 2016; Hoy et al., 2017).
Knowledge about the expected negative impact of increasing temperatures, via heat stress and
a reduction in the quality and quantity of resources, on the metabolic rate and phenotypic
plasticity of animals is therefore incomplete (Carey & Sigwart, 2014; Kruuk et al., 2015; Baar
et al., 2018; Riemer et al. 2018).

Furthermore, a decrease in the size of morphological traits has been associated with
selective hunting pressure (Coltman et al., 2003; Pérez et al., 2011; Douhard et al., 2016; Pigeon
et al. 2016); the directed removal of animals with specific traits. Again, the lack of sufficiently
long and well-replicated datasets complicates any systematic assessment of evolutionary
responses to intensive trophy harvesting (Büntgen et al., 2018; Festa-Bianchet & Mysterud,
2018). Hence, it remains debatable whether a properly managed hunting system can be used as
a conservation tool for maintaining wildlife biodiversity (Di Minin et al., 2016; Ripple et al.,
2016). Furthermore, unravelling the genetic and environmental mechanisms strong enough to
influence the morphology and/or phenology in animal populations, which may or may not be
contrasting (Quéméré et al., 2018), describes an important challenge in climate change-related
evolutionary ecology, with direct implications for conservation biology.

Here, we analyse a large collection of body weight, hind foot length and lower jaw length
measures of the four most abundant ungulate species in the European Alps. Our dataset includes
anatomical information from each animal of the two Bovid species ibex (*Capra ibex*) and

chamois (*Rupicapra rupicapra*), as well as the two Cervid species red deer (*Cervus elaphus*)
and roe deer (*Capreolus capreolus*), harvested between 1991 and 2013 in the Swiss Canton
Grisons (hereinafter GR). Shaping life-history traits and thereby influencing animal fitness,
~~both~~ skeletal measures can be used as indicators of juvenile growth and phenotypic expression
later in life (Albon et al., 1992; Clutton-Brock et al., 1982; Pettorelli et al., 2002); the result of
a complex interplay between an organism's genes and its environment. Large and heavy
individuals are expected to have higher lifetime reproductive success because big males should
gain access to more females (Clutton-Brock et al., 1992), and superior females should wean
more offspring (Clutton-Brock et al., 1988; Côte & Festa-Bianchet, 2001). Using different
regression functions and discrete timeseries analyses, as well as a more conservative split-
period approach, and accounting for possible effects of age, sex, phylogeny, biometric measure,
and harvest elevation, we test if the body weight and size of GR's wild ungulates has changed
since 1991 in response to increasing temperatures and persisting hunting pressure. Moreover,
we assess if males of the most dimorphic species were affected ~~stronger~~ than their female
counterparts ~~by warming-induced changes in their nutritional resources~~.

**Methods**

GR's hunting inventory includes 222,961 eviscerated body weight (EBW), 170,729 hind foot
length (HFL) and 145,980 lower jaw length (LJL) measurements of the sympatric ibex,
chamois, red deer and roe deer, harvested from 1991–2013 (Büntgen et al., 2017) (see
Supplementary for details). All of these measurements ~~have been~~ performed by official wildlife
wardens and gamekeepers following strict protocols. While EBW is more sensitive to
interannual changes in the animals'  environment, both skeletal measures – without removing
the animals' skin – are frequently used to characterize the growth of ungulates as a combined
result of genetic and ecological  factors. Each of the analysed 226.335 ungulates was living

within ~7,000 km² of heterogeneous landscape that comprises meadows, pastures, different
forests types, alpine tundra and cliffs, as well as permanent snow and ice fields, between 260
and 4,059 m asl. Permitted to local residents, the hunting of free-ranging chamois, red deer and
roe deer is restricted to two–three weeks in September; ibex to three weeks in October. No
trends and extremes in the weight, size and age of the harvested animals are inherent within and
between the species-specific autumnal hunting seasons (not shown). Moreover, trophy selection
and supplementary feeding of GR's wild ungulate populations is prohibited (Büntgen et al.,
2018).

In addition to the assessment of year-to-year changes in the species-specific sample size of
GR's harvested male and female ungulates (Supplementary Figure S1), we explored the
strength of any possible relationship between animals' age and their EBW, HFL, LJL. We also
analysed the yearly mean, median and standard deviation of EBW, HFL, LJL, and harvest
elevation, for each species, sex and age between 1991 and 2013. Additionally, we tested for
any interdependency of the three biometric measures, as well as for possible elevational effects
on the ungulates' body mass and size. All calculations were performed in R (R Core Team,
2018).

Finally, we computed the mean HFL and LJL for each species, sex and harvesting age over
two independent, though equally long subperiods from 1991–2000 and 2004–2013. While this
conservative approach ignores interannual variability, it provides a robust measure for longer-
term changes in the data mean values. Annual horn growth increments were used to age the
Bovids, whereas the condition of premolars and molars were considered for age estimates of
the Cervids. Acknowledging some degree of uncertainty in the age estimates of adult deer
(Hamlin et al., 2000), we clustered data for animals >3 years into four robust classes (i.e. 4–6,
7–9, 10–12, and 13+ years). Only individuals older than 1–16 months, hereafter referred to as
yearlings, were considered. The observed differences in body size between the two split periods

were validated by the non-parametric ‘Mann-Kendall’ and ‘Wilcoxon’ tests, as well as using
‘Holm-Bonferroni’ correlation coefficients. To quantify Europe’s unprecedented recent warming
trend (Schär et al., 2004), monthly temperature means, centred over the eastern Swiss Alps (10°
East and 46.5° North), were extracted from the latest generation of the ‘Gridded Berkeley Earth
Surface Temperature Anomaly Field’ (after Muller et al., 2013). Furthermore, we compared
monthly-resolved values of the normalized differenced vegetation index (NDVI), averaged from
1 km resolution grid boxes over GR’s study area (Büntgen et al., 2017), against the species-
specific year-to-year and longer-term trends in EBW, HFL, LJJ of the harvested animals.

**Results**

Year-to-year changes in EBW, HFL and LJJ of each species and sex indicate stable body mass
and size of GR’s four most abundant wild ungulate species since 1991 (Fig. 1). The generally
plastic and heritable measures jointly exhibit weak interannual variability, and there is no
statistical evidence for significant long-term trends in any of the data used. Independently
obtained from each age class (Table 1), this finding is supported by our split-period ~~approach~~
(Fig. 2). Irrespective of age, sex, phylogeny, and the biometric measure used, ungulate body
size in GR remained stable when independently calculated over two equally long early-late split
periods. None of the four species at any age shows a detectable reduction in EBW, HFL and
LJJ (Mann-Kendall $p > 0.05$ and t-test $p = 1$). **Regional mean annual surface temperature**
**anomalies, however, increased by 0.16°C in the same period of time (from 0.82°C in 1991–**
**2000 to 0.98°C in 2004–2013). while, the autumnal harvest intensity slightly decreased (from**
**108,856 to 89,238 harvested animals).**  nalysis further reveals that ibex, chamois and red
deer males are overall larger than same-aged females (Fig. 2), whereas roe deer body size of
both sexes is rather similar. The LJJ of male ibex and red deer is larger than that of females of
the same age, but tends to be nearly identical between male and female chamois and roe deer.

Both morphological traits reveal comparable growth patterns, with the smallest juvenile
increase in roe deer and chamois, and the most distinct and prolonged juvenile growth trend in
ibex and red deer. The sexual dimorphism ratio of HFL in adults is largest for ibex (1.15),
followed by chamois and red deer (both 1.06), and young males of all dimorphic species show
a generally faster increase in HFL than females (Fig. 2; Table 1). Statistically meaningful and
temporally stable relationships are not found between any of the animal parameters and the
high-resolution NDVI variables.

Moreover, our dataset reveals distinct, species- and sex-specific, relationships between
animal age and EBW, HFL and LJL (Supplementary Figures S2–S4). Due to allometric
constraints and growth priorities, there are positive associations between the animals' more
plastic EBW (a compound between body size and condition, with a particularly strong influence
of size on the mass of young individuals) and the generally more heritable HFL and LJL
(Supplementary Figures S5–S6). Likewise, there is a tendency for stronger linkages between
the mass and size of younger and smaller animals compared to older and larger specimens. A
slightly more complex picture is obtained from comparisons of the three biometric measures of
GR's ungulate species against their harvest elevation (Supplementary Figures S7–S10). The
EBW and LJL of both, ibex and chamois males and females are positively related to their
harvest elevation, whereas this overall positive relationship is less pronounced for the Bovids'
HFL. Generally hunted at lower elevations, the body mass and size of red deer tends to be less
dependent on harvest elevation.

48
49 172

51 173 **Discussion and Conclusions**

Unlike previous work (Yom-Tov, 2001; Gardner et al., 2009, 2011, 2014; Ozgul et al., 2009;
Sheridan & Bickford, 2011; Rughetti et al., 2012; Gibert & DeLong, 2014; Legagneux et al.,
2014; Mason et al., 2014; Caruso et al., 2015; van Gils et al., 2016; Horne et al., 2017; Hoy et

al., 2017; MacLean & Beissinger, 2017; Tseng et al., 2017; Lindmark et al., 2018), and despite
increasing temperatures (Schär et al., 2004) and continuous harvest practise (Büntgen et al.,
2017), this study suggests stable body mass and size of GR's wild ungulates between 1991 and
2013. Due to a sustainable management of natural resources, the season, strategy and intensity
of the hunting system did not change during the study period, and species-specific population
sizes remained approximately stable (Büntgen et al., 2017, 2018). Exhibiting similar results,
the longer HFL that range from ~28–54 cm and the shorter LJL measurements that range from
~16–32 cm (both recorded at a resolution of 1.0 millimetre), are indicative for the accuracy of
the herein analysed biometric parameters (Martin et al., 2013). Since HFL and LJL are age-
specific and easy to measure, they have advantages over EBW, which may exhibits more intra-
and interannual variability due to environmental-induced changes in the animals' nutritional
constitution (Toigo et al., 2006). In agreement with Becciolini et al., (2016), the HFL and LJL
of GR's red deer population mainly increase during juvenile. In contrast to Pettorelli et al.,
(2005, 2017), our study does not reveal any direct effects of NDVI on interannual to decadal
changes in the weight and size of GR's most abundant ungulate species since 1991. Moreover,
our findings of juvenile chamois also contrast the results of Mason et al., (2014), who reported
shrinking juvenile body mass in three neighbouring chamois populations in the Italian Alps due
to indirect environmental effects on resource productivity and phenology.

Possible reasons for the lack of detectable positive long-term changes in both, the more
plastic as well as more heritable traits are manifold. Despite the fact that our study was
conducted during Europe's warmest period of the past two millennia (Luterbacher et al., 2016),
the observational period might still be too short, and/or with too little warming in-between
(Medhaug et al., 2017), or because evolutionary and phenotypic responses, in contrast to
environmental effects (Rughetti et al., 2012; Büntgen et al., 2014; Quéméré et al., 2018), are
particularly slow in species with long generation times (Merilä et al., 2012). Due to relatively

low reproduction and harvesting ages, and thus the species' overall high turnover rate, we
expect detectable changes within a few decades. Since we analysed harvest data from animals
that inhabit lower, intermediate or higher altitude areas, temporal changes in the elevational
movement of ungulates may mask some of the expected responses (Büntgen et al., 2017). In
addition, positive environmental factors such as more and better nutrition supply during longer
and warmer vegetation periods may counterbalance some of the negative effects of recent
warming and selective hunting on the body size of Alpine ungulates (Yom-Tov & Geffen,
2011). Since the quality and quantity of forage is jointly determined by climate variation and
grassland management, including the widespread use of fertilizers (Donald et al., 2001), their
covariance may blur a direct negative relationship, between increasing ambient temperature and
shrinking body size due to thermoregulatory efficiency. Although our findings are independent
of sex and species (selection pressure should be largest for ibex and red deer males but lowest
for roe deer females), some additional bias may originate from the general tendency of targeting
larger animals within a given age-class (Büntgen et al., 2018), and warming-induced elevational
range shifts (Büntgen et al., 2017).

Phenotypic plasticity of behavioural, physiological and morphological traits enables
organisms to cope with environmental changes based on the natural history they experienced
after the Pleistocene (Davis et al., 2005). Such changes are considered to impose selection
pressure on traits important for fitness (Yom-Tov & Yom-Tov, 2004; Quéméré et al., 2018),
such as body size. Our findings, however, contradict significant long-term changes in the annual
mean values of EBW, HFL and LJJ in any of the four species. This suggests that the responsible
mechanism of these traits has remained unchanged during the study period, and corresponds
with studies on *Ovis canadensis* (Coltman et al., 2005) and *Cervus elaphus* (Kruuk et al., 2000),
in which selection has not depleted additive genetic variance in morphological traits. Skeletal
structures particularly depend on the allocation of resources for growth during the juvenile life

stage (Hou et al., 2008). At the adult stage, skeletal size remains stable in most endotherms
(Yom-Tov & Geffen, 2011), and agrees with data presented here. Nevertheless, the magnitude
of environmental drivers, such as temperature variability and food availability, should not be
the same for all four species in this study due to species-specific ecological requirements
(Büntgen et al., 2017). Heat dissipation by the smaller roe deer, for instance, should be better
than the ibex or red deer. On the other hand, a better surface to size ratio is beneficial for
sustaining cold winter seasons that last longer than the snow-free summers in Alpine
environments (Ernakovich et al., 2014). Varying in space and time, these opposing drivers may
counterbalance directional selection of morphometric growth traits in the studied species.
Moreover, GR's complex topography allows individuals to migrate at intra- and interannual
timescales (Davis et al., 2005; Büntgen et al. 2017), thus mitigating a possible climate-induced
selection pressure.

33 240 **Acknowledgements**

We are very thankful to the numerous hunters, wildlife wardens and gamekeepers who
contributed enthusiastically to the development of Grisons' long-term hunting inventory. Marco
Festa-Bianchet and Atle Mysterud kindly commented on earlier versions of this manuscript.

44 245 **Funding**

This study was partly supported by the SustES project (CZ.02.1.01/0.0/0.0/16_019/0000797)
of the Czech Republic.

53 249 **Author contributions**

UB conceived and designed the research, analysed the data and wrote the article. HJ compiled

the data and provided critical input. JDG and AP analysed the data and contributed to
discussion. PJK and KB co-wrote the article. All authors contributed to discussion and approved
submission.

**References**

ALBON, S.D., CLUTTON-BROCK, T.H. & LANGVATN, R. (1992) *Cohort variation in reproduction*
*and survival: Implications for population demography*. In: Brown RD (ed) *The Biology of*
*Deer*. Springer Verlag, Berlin.
BAAR, Y., FRIEDMAN, A.L.L., MEIRI, S. & SCHARF, I. (2018) Little effect of climate change on
body size of herbivorous beetles. *Insect Science*, 25, 309–316.
BECCIOLINI, V., BOZZI, R., VILIANI, M., BIFFANI, S. & PONZETTA, M.P. (2016) Body
measurements from selective hunting: biometric features of red deer (*Cervus elaphus*) from
Northern Apennine, Italy. *Italian Journal of Animal Sciences*, 15, 461–472.
BERGMANN, C. (1847) *Über die Verhältnisse der Wärmeökonomie der Thiere zu ihrer Größe*.
*Göttinger Studien* 3, 595–708.
BÜNTGEN, U., JENNY, H., LIEBHOLD, A., MYSTERUD, A., EGLI, S., NIEVERGELT, D., STENSETH,
267 N.C. & BOLLMANN, K. (2014) European springtime temperature synchronizes ibex horn
growth across the eastern Swiss Alps. *Ecology Letters*, 17, 303–313.
BÜNTGEN, U., GREUTER, L., BOLLMANN, K., JENNY, H., LIEBHOLD, A., GALVAN, J.D.,
STENSETH, N.C., ANDREW, C. & MYSTERUD, A. (2017) Elevational range shifts in four
mountain ungulate species from the Swiss Alps. *Ecosphere*, 8, e01761.
BÜNTGEN, U., GALVAN, J.D., MYSTERUD, A., KRUSIC, P.J., HÜLSMANN, L., JENNY, H., SENN, J.
& BOLLMANN, K. (2018) Horn growth variation and hunting selection of the Alpine ibex.
*Journal of Animal Ecology*, 87, 1069–1079.

CAREY, N. & SIGWART, J.D. (2014) Size matters: plasticity in metabolic scaling shows body-
size may modulate responses to climate change. *Biology Letters*, 10, 20140408.
CARUSO, N.M., SEARS, M.W, ADAMS, D.C. & LIPS, K.R. (2015) Widespread rapid reductions
in body size of adult salamanders in response to climate change. *Global Change Biology*,
20, 1751–1759.
CLUTTON-BROCK, T.H., GUINNESS, F.E. & ALBON, S.D. (1982) *Red deer: Behavior and ecology*
*of two sexes*. University of Chicago Press.
CLUTTON-BROCK, T.H., ALBON, S.D. & GUINNESS, F.E. (1988) *Reproductive success in male*
*and female red deer*. In: Clutton-Brock, T.H. (ed) *Reproductive success*. University of
Chicago Press.
CLUTTON-BROCK, T.H., PRICE, O.F. & MACCOLL, A.D.C. (1992) Mate retention, harassment,
and the evolution of ungulate leks. *Behavioural Ecology*, 3, 234–242.
COLTMAN, D.W., O'DONOGHUE, P., JORGENSON, J.T., HOGG, J.T., STROBECK, C. & FESTA-
BIANCHET, M. (2003) Undesirable evolutionary consequences of trophy hunting. *Nature*,
426, 655–658.
COLTMAN, D.W., DONOGHUE, P.O., HOGG, J.T. & FESTA-BIANCHET, M. (2005) Selection and
genetic (co) variance in bighorn sheep. *Evolution*, 59, 1372–1382.
CÔTE, S.D. & FESTA-BIANCHET, M. (2001) Birthdate, mass and survival in mountain goat kids:
effects of maternal characteristics and forage quality. *Oecologia*, 127, 230–238.
DARWIN, C. (1859) *On the Origin of Species*. London: John Murray 502.
DAVIS, M.B., SHAW, R.G. & ETTERTSON, J.R. (2005) Evolutionary responses to changing
climate. *Ecology*, 86, 1704–1714.
DI MININ, E., LEADER-WILLIAMS, N. & BRADSHAW, C.J.A. (2016) Banning trophy hunting will
exacerbate biodiversity loss. *Trends in Ecology and Evolution*, 31, 99–102.

DONALD, P.F., GREEN, R.E. & HEATH, M.F. (2001) Agricultural intensification and the collapse
of Europe's farmland bird populations. *Proceedings of the Royal Society B*, 268, 25–29.
DOUHARD, M., FESTA-BIANCHET, M., PELLETIER, F., GAILLARD, J.-M. & BONENFANT, C. (2016)
Changes in horn size of Stone's sheep over four decades correlate with trophy hunting
pressure. *Ecological Applications*, 26, 309–321.
DOUHARD, M., PIGEON, G., FESTA-BIANCHET, M., COLTMAN, D.W., GUILLEMETTE, S. &
PELLETIER, F. (2017) Environmental and evolutionary effects on horn growth of male
bighorn sheep. *Oikos*, 126, 1031–1041.
ERNAKOVICH, J.G., HOPPING, K.A., BERDANIER, A.B., SIMPSON, R.T., KACHERGIS, E.J.,
STELTZER, H. & WALLENSTEIN, M.D. (2014) Predicted responses of arctic and alpine
ecosystems to altered seasonality under climate change. *Global Change Biology*, 20, 3256–
3269.
FESTA-BIANCHET, M. & MYSTERUD, A. (2018) Hunting and evolution: theory, evidence, and
unknowns. *Journal of Mammalogy*, 99, 1281–1292.
GARDNER, J.L., HEINSOHN, R. & JOSEPH, L. (2009) Shifting latitudinal clines in avian body size
correlate with global warming in Australian passerines. *Proceedings of the Royal Society B*,
276, 3845–3852.
GARDNER, J.L., PETERS, A., KEARNEY, M.R., JOSEPH, L. & HEINSOHN, R. (2011) Declining body
size: a third universal response to warming? *Trends Ecology and Evolution*, 26, 285–291.
GARDNER, J.L., AMANO, T., BACKWELL, P.R.Y., IKIN, K., SUTHERLAND, W.J. & PETERS, A.
(2014) Temporal patterns of avian body size reflect linear size responses to broadscale
environmental change over the last 50 years. *Journal of Avian Biology*, 45, 529–535.
GIBERT, J.P. & DELONG, J.P. (2014) Temperature alters food web body-size structure. *Biology*
*Letters*, 10, 20140473.

HAMLIN, L.L., PAC, D.F., SIME, C.A., DE SIMONE, R.M. & DUSEK, G.L. (2000) Evaluating the
accuracy of ages obtained by two methods for Montana ungulates. *The Journal of Wildlife*
*Management*, 64, 441–449.
HORNE, C.R., HIRST, A.G. & ATKINSON, D. (2017) Seasonal body size reductions with warming
covary with major body size gradients in arthropod species. *Proceedings of the Royal*
*Society B*, 284, 20170238.
HOU, C., ZUO, W., MOSES, M.E., WOODRUFF, W.H., BROWN, J.H. & WEST, G.B. (2008) Energy
uptake and allocation during ontogeny. *Science*, 322, 736–739.
HOY, S.R., PETERSON, R.O. & VUCETICH, J.A. (2017) Climate warming is associated with
smaller body size and shorter lifespans in moose near their southern range limit. *Global*
*Change Biology*, 24, 2488–2497.
KRUUK, L.E.B., CLUTTON-BROCK, T.H., SLATE, J., PEMBERTON, J.M., BROTHERSTONE, S. &
GUINNESS, F.E. (2000) Heritability of fitness in a wild mammal population. *Proceedings of*
*the National Academy of Science USA*, 97, 698–703.
KRUUK, L.E.B., OSMOND, H.L. & COCKBURN, A. (2015) Contrasting effects of climate on
juvenile body size in a Southern Hemisphere passerine bird. *Global Change Biology*, 21,
2929–2941.
LEGAGNEUX, P., GAUTHIER, G., LECOMTE, N., SCHMIDT, N.M., REID, D., CADIEUX, M.-B.,
BERTEAUX, D., BÉTY, J., KREBS, C.J., IMS, R.A., YOCCOZ, N.G., MORRISON, R.I.G., LEROUX,
S.J., LOREAU, M. & GRAVEL, D. (2014) Arctic ecosystem structure and functioning shaped
by climate and herbivore body size. *Nature Climate Change*, 4, 379–383.
LINDMARK, M., HUSS, M., OHLBERGER, J. & GRÅDMARK, A. (2018) Temperature-dependent
body size effects determine population responses to climate warming. *Ecology Letters*, 21,
181–189.

LUTERBACHER, J., ET AL. (2016) European summer temperatures since Roman times.
*Environmental Research Letters*, 11, 024001.
MACLEAN, S.A. & BEISSINGER, S.R. (2017) Species' traits as predictors of range shifts under
contemporary climate change: A review and meta-analysis. *Global Change Biology*, 23,
4094–4105.
MARTIN, J.G.A., FESTA-BIANCHET, M., CÔTÉ, S.D. & BLUMSTEIN, D.T. (2013) Detecting
between-individual differences in hind-foot length in populations of wild mammals.
*Canadian Journal of Zoology*, 91, 118–123.
MASON, T.H.E., APOLLONIO, M., CHIRICHELLA, R., WILLIS, S.G. & STEPHENS, P.A. (2014)
Environmental change and long-term body mass declines in an alpine mammal. *Frontiers*
*in Zoology*, 11, 69.
MEDHAUG, I., STOLPE, M.B., FISCHER, E.M. & KNUTTI, R. (2017) Reconciling controversies
about the 'global warming hiatus'. *Nature*, 545, 41–47.
MERILÄ, J. (2012) Evolution in response to climate change: in pursuit of the missing evidence.
*BioEssays*, 34, 811–818.
MULLER, R.A., CURRY, J., GROOM, D., JACOBSEN, R., PERLMUTTER, S., ROHDE, R., ROSENFELD,
363 A., WICKHAM, C. & WURTELE, J. (2013) Decadal variations in the global atmospheric land
temperatures. *Journal of Geophysical Research A*, 118, 1–7.
OZGUL, A., TULJAPURKAR, S., BENTON, T.G., PEMBERTON, J.M., CLUTTON-BROCK, T.H. &
COULSON, T. (2009) The dynamics of phenotypic change and the shrinking sheep of St.
Kilda. *Science*, 325, 464–467.
PÉREZ, J.M., SERRANO, E., GONZÁLEZ-CANDELA, M., LEÓN-VIZCAINO, L., BARBERÁ, G.G., DE
SIMÓN, M.A., FANDOS, P., GRANADOS, J.E., SORIGUER, R.C. & FESTA-BIANCHET, M. (2011)
Reduced horn size in two wild trophy-hunted species of Caprinae. *Wildlife Biology*, 17,
102–112.

PETTORELLI, N., GAILLARD, J.-M., VAN LAERE, G., DUNCAN, P., KJELLANDER, P., LIBERG, O.,
DELMORE, D. & MAILLARD, D. (2002) Variations in adult body mass in roe deer: The effects
of population density at birth and of habitat quality. Intense selective hunting leads to
artificial evolution in horn size. *Proceedings of the Royal Society B*, 269, 747–753.
PETTORELLI, N., VIK, J.O., MYSTERUD, A., GAILLARD, J.-M., TUCKER, C.J. & STENSETH, N.C.
(2005) Using the satellite-derived NDVI to assess ecological responses to environmental
change. *Trends in Ecology and Evolution*, 20, 503–510.
PETTORELLI, N., ET AL. (2017) Satellite remote sensing of ecosystem functions: opportunities,
challenges and way forward. *Remote Sensing in Ecology and Conservation*, 4, 71–93.
PIGEON, G., FESTA-BIANCHET, M., COLTMAN, D.W. & PELLETIER, F. (2016) Intense selective
hunting leads to artificial evolution in horn size. *Evolutionary Applications*, 9, 521–530.
QUÉMÉRÉ, E., GAILLARD, J.-M., GALAN, M., VANPÉ, C., DAVID, I., PELLERIN, M., KJELLANDER,
P., HEWISON, A.J.M. & PEMBERTON, J.M. (2018) Between-population differences in the
genetic and maternal components of body mass in roe deer. *BMC Evolutionary Biology*, 18,
39.
R CORE TEAM. (2018) *R: A language and environment for statistical computing*. R Foundation
for Statistical Computing, Vienna, Austria.
RIEMER, K., ANDERSON-TEIXEIRA, K.J., SMITH, F.A., HARRIS, D.J. & ERNEST, S.K.M. (2018)
Body size shifts influence effects of increasing temperatures on ectotherm metabolism.
*Global Ecology and Biogeography*, 27, 958–967.
RIPPLE, W.J., NEWSOME, T.M. & KERLEY, G.I. (2016) Does trophy hunting support
biodiversity? A response to Di Minin et al. *Trends in Ecology and Evolution*, 31, 495–496.
RUGHETTI, M. & FESTA-BIANCHET, M. (2012) Effects of spring–summer temperature on body
mass of chamois. *Journal of Mammalogy*, 93, 1301–1307.

SCHÄR, C., VIDALE, P.L., LÜTHI, D., FREI, C., HÄBERLI, C., LINIGE, M.A. & APPENZELLER, C.
(2004) The role of increasing temperature variability in European summer heatwaves.
*Nature*, 427, 332–336.
SHERIDAN, J.A. & BICKFORD, D. (2011) Shrinking body size as an ecological response to
climate change. *Nature Climate Change*, 1, 401–406.
- TOÏGO, C., GAILLARD, J.-M., VAN LAERE, G., HEWISON, M. & MORELLET, N. (2006) How does
environmental variation influence body mass, body size, and body condition? Roe deer as
a case study. *Ecography*, 29, 301–308.
- TSENG, M., KAUR, K.M., SOLEIMANI PARI, S., SARAI, K., CHAN, D., YAO, C.H., PORTO, P.,
TOOR, A., TOOR, H.S. & FOGRASCHER, K. (2017) Decreases in beetle body size linked to
climate change and warming temperatures. *Journal of Animal Ecology*, 87, 647–659.
- VAN GILS, J.A., LISOVSKI, S., LOK, T., MEISSNER, W., OŻAROWSKA, A., DE FOUW, J.,
RAKHIMBERDIEV, E., SOLOVIEV, M.Y., PIERSMA, T. & KLAASSEN, M. (2016) Body
shrinkage due to Arctic warming reduces red knot fitness in tropical wintering range.
*Science*, 352, 819–821.
- 37 411 YOM-TOV, Y. (2001) Global warming and body mass decline in Israeli passerine birds.
38
39 412 *Proceedings of the Royal Society B*, 268, 947–952.
- YOM-TOV, Y. & YOM-TOV, S. (2004) Climatic change and body size in two species of Japanese
rodents. *Biological Journal of the Linnean Society*, 82, 263–267.
- YOM-TOV, Y. & GEFFEN, E. (2011) Recent spatial and temporal changes in body size of
terrestrial vertebrates: probable causes and pitfalls. *Biological Reviews*, 86, 531–541.

TABLE 1. Comparison of long-term mean and variance changes in eviscerated body weight (EBW), hind foot length (HFL) and lower jaw length (LJL) of Grisons' wild ungulates between the two subperiods 1991–2000 and 2004–2013. While the first value always refers to the temporal average, the second values indicates the corresponding standard deviation. Most importantly, none of the 192 pairings reveals a significant difference between the early and late interval.

		1991-2000 (all)	2004-2013 (all)	1991-2000 (1yr)	2004-2013 (1yr)	1991-2000 (2-4yrs)	2004-2013 (2-4yrs)	1991-2000 (>8yrs)	2004-2013 (>8yrs)
Ibex	EBW Female	26.49 ±6.06	27.17 ±6.09	16.89 ±2.67	17.26 ±2.56	25.29 ±3.96	25.96 ±3.99	30.68 ±4.18	31.18 ±4.16
	Male	46.54 ±17.37	47.24 ±7.1	17.56 ±3.52	18.93 ±3.57	36.28 ±8.21	35.83 ±8.27	68.20 ±8.64	67.41 ±8.45
	HFL Female	29.04 ±1.38	29.12 ±.43	27.25 ±1.17	27.39 ±1.09	29.05 ±1.15	29.11 ±1.25	29.65 ±1.03	29.71 ±1.03
	Male	32.68 ±2.04	32.74 ±1.93	28.11 ±1.45	28.36 ±1.26	32.26 ±1.65	32.16 ±1.51	33.94 ±1.17	34.04 ±1.15
	LJL Female	29.04 ±1.47	29.12 ±1.63	15.11 ±0.81	14.96 ±0.72	17.05 ±0.95	17.34 ±1.06	18.6 ±.75	18.61 ±0.71
	Male	32.68 ±1.62	32.74 ±1.45	15.51 ±0.98	16.95 ±1.19	18.11 ±1.18	18.28 ±1.08	20.03 ±0.81	20.05 ±0.78
Chamois	EBW Female	19.44 ±3.66	18.91 ±3.57	15.02 ±2.29	15.28 ±2.18	19.98 ±2.67	19.90 ±2.47	21.23 ±2.73	21.09 ±2.56
	Male	25.58 ±5.99	25.05 ±5.94	15.34 ±2.35	15.36 ±2.41	26.41 ±3.81	25.48 ±3.87	28.83 ±3.27	28.56 ±3.23
	HFL Female	33.79 ±1.43	33.82 ±1.35	32.66 ±1.42	32.93 ±1.34	34.09 ±1.23	34.16 ±1.08	34.19 ±1.17	34.35 ±1.09
	Male	35.43 ±1.66	35.45 ±.63	33.46 ±1.49	33.52 ±1.55	35.80 ±1.26	35.75 ±1.22	35.97 ±1.43	36.73 ±1.21
	LJL Female	14.99 ±0.95	15.02 ±.04	13.78 ±0.64	14.02 ±0.79	15.04 ±0.65	15.33 ±0.57	15.63 ±0.61	15.69 ±0.61
	Male	15.34 ±0.97	15.19 ±0.83	13.87 ±0.67	13.8 ±0.56	15.51 ±0.66	15.31 ±0.55	15.86 ±0.66	15.66 ±0.49
Roe deer	EBW Female	16.37 ±2.07	16.03 ±2.12	15.56 ±1.76	15.46 ±1.63	16.9 ±1.80	16.94 ±1.63	17.08 ±1.95	16.97 ±1.95
	Male	17.48 ±2.05	17.43 ±2.04	15.98 ±1.72	16.14 ±1.68	18.02 ±1.71	18.11 ±1.63	17.97 ±1.84	17.99 ±2.03
	HFL Female	34.55 ±1.29	34.5 ±1.30	34.35 ±1.17	34.40 ±1.04	34.8 ±1.07	34.87 ±1.02	34.89 ±1.02	35.06 ±0.96
	Male	34.94 ±1.28	34.94 ±1.16	34.73 ±1.21	34.73 ±1.09	35.07 ±1.20	35.13 ±1.05	35.4 ±0.86	35.04 ±1.16
	LJL Female	15.34 ±0.68	15.24 ±0.63	15.02 ±0.57	14.99 ±0.48	15.53 ±0.57	15.46 ±0.48	15.74 ±0.58	15.59 ±0.54
	Male	15.48 ±0.65	15.39 ±0.56	15.05 ±0.60	15.01 ±0.49	15.61 ±0.56	15.54 ±0.48	15.76 ±0.67	15.62 ±0.47
Red deer	EBW Female	61.58 ±11.17	60.16 ±11.55	55.98 ±6.71	54.69 ±6.46	65.71 ±7.39	64.84 ±7.21	73.44 ±8.29	73.40 ±8.41
	Male	90.03 ±24.87	89.04 ±25.31	63.17 ±8.88	61.63 ±8.95	91.37 ±14.15	88.78 ±13.19	133.63 ±19.58	135.52 ±20.91
	HFL Female	48.35 ±2.38	48.22 ±2.27	47.80 ±20.6	47.77 ±1.83	48.93 ±1.95	48.86 ±1.83	49.75 ±1.88	49.57 ±1.79
	Male	51.1 ±2.77	51.12 ±2.62	49.3 ±2.23	49.22 ±2.34	51.62 ±2.15	51.61 ±1.94	52.8 ±1.98	52.74 ±2.04
	LJL Female	25.81 ±2.29	25.96 ±2.36	24.43 ±1.15	24.77 ±1.23	26.79 ±1.22	27.06 ±1.26	28.44 ±1.15	28.75 ±1.14
	Male	27.85 ±2.49	28.2 ±2.51	25.28 ±1.44	25.53 ±1.45	28.35 ±1.32	28.66 ±1.35	30.76 ±1.46	31.07 ±1.46

 FIG. 1. Year-to-year (vertical bars from 1991–2013) and long-term (horizontal lines of the two
 subperiods 1991–2000 and 2004–2013) changes in eviscerated body weight (EBW), hind foot
 length (HFL) and lower jaw length (LJJ) of Grisons' wild ungulates. Box plots show the
 distribution range of all data per year and age, with horizontal white lines representing the
 median, and the upper/lower box limits referring to the 75th/25th percentile. Whiskers extend
 from the two ends of the box until they reach the sample maximum and minimum. Outliers are
 not shown to facilitate the reading. Horizontal lines show the species-specific, subperiod grand
 average of all harvested male and female ungulates, and y-axes have slightly different scales to
 facilitate visual comparison. Performing the same analysis for each age class (4 x 2 x 7) resulted
 in 56 cases that showed no differences between the two subperiods (see Fig. 2, Table 1 for
 details).

 FIG. 2. Long-term changes in hind foot length (HFL) and lower jaw length (LJL) of Grisons'
 wild ungulates between the two subperiods 1991–2000 and 2004–2013. Vertical lines indicate
 standard errors and y-axes have slightly different scales to facilitate visual comparison (see
 Supplementary for details).

UNIVERSITY OF
CAMBRIDGE

Department of Geography

Appendix C

Professor Matthew Collins
Associate Editor *Royal Society Open Science*
University of Cambridge, UK

Professor Ulf Büntgen
Professor of Environmental Systems Analysis

20th May 2020

Dear Prof Collins, dear Editors

Please find attached our carefully revised manuscript “**Stable body size of Alpine ungulates**”, which we hope is now suitable for publication in *RSOS*.

We are thankful to both referees for their overall positive and encouraging evaluation of our work. Together with the valuable editorial comments, we considered all minor and major suggestions and improved the article accordingly.

All changes are described in a separate point-by-point response letter.

Please do not hesitate contacting me in case anything remains unclear and/or further information is needed.

We kindly ask you to consider this manuscript for publication in *RSOS* and look forward to hearing from you soon.

On behalf of the authors;
Yours respectfully

Ulf Büntgen

Downing Place
Cambridge CB2 3EN
Tel: +44 (0) 1223 760564
Fax: +44 (0) 1223 333392
Email: ulf.buentgen@geog.cam.ac.uk

Appendix D

Point-by-point response letter

Associate Editor's comments (Professor Matthew Collins) Both reviewers believe that the MS is a worthwhile publication, but both suggest major reviews.

We carefully considered all of their comments and suggestions and revised the manuscript accordingly.

The primary concern of Reviewer#1 was the statistical framework and suggests using multiple regressions instead. I would ask you to consider this suggestion. Please note that if you choose not to adopt the strategy proposed by Reviewer#1, and given that this may be raised by other readers, I think that it would also be useful to include this justification in your methods section as well as in your response to me.

We performed a multiple regression analysis for each of the four species and both sexes, added new text bits to both Methods and Results, and included four corresponding Tables into the supplementary materials (Table S1–S4).

Both Reviewer#1 and Reviewer#2 would also like more information on the spatial and topographic variation in the dataset.

We now better describe the spatial and topographic aspects of the dataset, and emphasize this issue prominent at the end of the abstract and in the conclusions. Moreover, we refer explicitly to our previous study (Büntgen et al. 2017), which focussed on climate-induced elevational range shifts of the four main Alpine ungulate species at intra- and interannual time-scales during the past decades.

*Büntgen U, Greuter L, Bollmann K, Jenny H, Liebhold A, Galvan JD, Stenseth NC, Andrew C, Mysterud A (2017) Elevational range shifts in four mountain ungulate species from the Swiss Alps. *Ecosphere* 8(4): e01761. 10.1002/ecs2.1761*

I have added further editorial comments to the PDF supplied by Reviewer #2 which you should find attached. (RSOS-200196_Proof_hi ijj.pdf)

We kindly acknowledge your valuable support, and considered all comments and suggestions together with those of the referee(s).

Reviewer#1

I reviewed the manuscript entitled “Stable body size of Alpine ungulates” and I found the problem raised in the manuscript both interesting and important. In my opinion it will be a valuable contribution to Royal Society Open Science as the manuscript is well written and structured.

Many thanks for this overall positive and encouraging evaluation of our study.

However, I found problems with the applied statistical approach. In my opinion, the statistical framework should be changed to make results clearer and easier to follow. Instead of running multiple tests on different subsets of data I propose to build separate models (multiple regressions) for every studied species with EBW, HFL LJJ as dependent variables and the year, age, altitude, NDVI value as independent variables.

We performed a multiple regression analysis for each of the four species and both sexes, added new text bits to both Methods and Results, and included four corresponding Tables into the supplementary materials (Table S1–S4).

The authors did not mention anything about spatial distribution of sampling that can induce spatial autocorrelation in model residuals. Unfortunately, many statements in the results section (L 110-112.) are not supported by any statistical testing.

We now better refer to intra- and interannual changes in harvest elevation, and re-wrote parts of the Results section.

I do not understand the reasoning provided in this sentence. Fig S2, S3, S4. Why the difference in the age of roe deer given in 1-year intervals, when in methods and in Fig 2 you provide that you estimated age for this species in broader age classes?

We re-wrote the description of age classes “Acknowledging some degree of uncertainty in the age estimates of adult deer (Hamlin et al., 2000), we clustered our dataset into the following age classes: 1, 2, 3, 4–6, 7–9, 10–12, and 13+ years (only individuals over 15–16 months are considered as yearlings).”

Reviewer #2

The impact of changes in climatological conditions on animal morphology, physiology, behaviour and life history requires the analysis of continuous, large, long-term data sets. These are particularly hard to find for large mammals. In this manuscript, the authors analyse long-term datasets for 4 artiodactyl species from a relatively small canton in the Swiss alps. The time period for the data runs from 1991–2013 with the data on body weight and morphology derived from animals culled in apparently well proscribed methodologies over this period. In the analysis the authors control for age, sex across the time period of the analyses.

Correct.

They find that there is no apparent change in weight of size of any of the species, ages or sex class across the period of investigation. This is a robust result which agrees with some other studies and disagrees with others.

Correct.

The authors conclude that the species studied in this context are able to move across landscapes (a mosaic of different land use and habitat types) and elevation in the light of changes in climatic conditions.

Correct.

This is an important and valuable study because of the size of the dataset, across 4 artiodactyl species. The manuscript is generally well written, although there are areas that require

clarification and a number of editorial changes that should be made to English (see attached PDF (RSOS-200196_Proof_hi ijg.pdf)).

Many thanks, the English has been improved during revision.

There is one area of improvement that can be made, that is that the topography of the region apparently allows the ungulates to ameliorate the effects of temperature increases over the study period by changing elevation and most probably habitat/land use type use. The authors should present an analysis of the interaction between elevation and year on body weight and size for males and females of different ages in their 4-study species. This would allow them to assess whether there is an effect within and across elevation over time.

We now better describe the spatial and topographic aspects of the dataset, and emphasize this issue prominent at the end of the abstract and in the conclusions. Moreover, we refer explicitly to our previous study (Büntgen et al. 2017), which focussed on climate-induced elevational range shifts of the four main Alpine ungulate species at intra- and interannual time-scales during the past decades.

Büntgen U, Greuter L, Bollmann K, Jenny H, Liebhold A, Galvan JD, Stenseth NC, Andrew C, Mysterud A (2017) Elevational range shifts in four mountain ungulate species from the Swiss Alps. Ecosphere 8(4): e01761. 10.1002/ecs2.1761

Specific (also see the attached PDF for edits to the manuscript)

All comments and suggestions provided in the PDF have been considered carefully.

I. 34-35. Latin names for species as well please

Added.

I. 53-54. Why is it an "important task for conservation managers"? This is a very presumptive statement. I can understand that it is scientifically interesting and may be important for sport hunters but conservation managers?

Because conservation managers should be concerned if global warming affects the body mass and size of animals. We slightly changed the sentence to "Understanding climate-induced changes in the weight and size of species is therefore an important task for scientists and conservation managers around the world."

I. 55. See also recent work on nightingales. <https://academic.oup.com/auk/advance-article-abstract/doi/10.1093/auk/ukaa012/5814682>

We added the citation and adapted the reference list accordingly.

REMACHA, C., RODRÍGUEZ, C., DE LA PUENTE, J., & PÉREZ-TRIS, J. (2020) Climate change and maladaptive wing shortening in a long-distance migratory bird. The Auk: Ornithological Advances, 137, 1–15.

I. 61 When would "Knowledge" be "complete"?

Changed to "The expected negative impacts of increasing temperatures, via heat stress and a reduction in the quality and quantity of resources, on the metabolic rate and phenotypic

plasticity of animals are not well understood (Carey & Sigwart, 2014; Kruuk et al., 2015; Baar et al., 2018; Riemer et al. 2018).".

I. 68. I am not sure how changes in weight/size with climate links to the management of biodiversity in general; maybe gene frequencies in a population/species.

We replaced "wildlife biodiversity" by "sustainability".

I. 70-73. A complex sentence; breakdown & redraft

We simplified the sentence "Unravelling the genetic and environmental mechanisms strong enough to influence the morphology and/or phenology in animal populations describes another important challenge (Quéméré et al., 2018).".

I. 99-101. References require for statements wrt the interannual variation in weight & the genetic/environmental condition on skeletal characteristics.

We slightly re-wrote the sentence and added references "Despite some degree of uncertainty (Serrano et al., 2008), EBW is sensitive to annual changes in the animals' environment (Mason et al., 2014), whereas both skeletal measures – taken without removing the animals' skin – are useful to characterize the growth of ungulates as a combined result of genetic and ecological factors (Toïgo et al., 2006).".

L. 104-105. Is "local resident" hunting different from the hunting by wardens? I assume that the records from the "permitted" hunts are not included in the dataset.

Changed to "Permitted to the residents and wardens of GR,".

I. 124-125. Why were these age groupings chosen?

Changed to "Acknowledging some degree of uncertainty in the age estimates of adult deer (Hamlin et al., 2000), we clustered our dataset into the following age classes: 1, 2, 3, 4–6, 7–9, 10–12, and 13+ years (only individuals over 15–16 months are considered as yearlings).".

I. 125. "Only individuals older than 15-16 months were considered" but you have just said you clustered "animals > 3 year.

See previous response.

I. 145-148. Move this to the beginning of the Results to provide climatological background.

Done.

I. 159-171. This is of limited importance for the manuscript. Reduce substantially (mainly the elevation effect should remain [see comment in General above]).

We moved this section to the "Methods", where we now better describe the dataset.

I. 180. How do you know that the hunting system is "sustainable"?

Changed to "GR's persistent ...".

I. 201. Or the effects on the weight/size of the species have already happened and the species have reached it adaptive trough wrt weight & size?

Many thanks, we added this argument to the Discussion.

I. 210. Is the increase in fertiliser happening in this context? Wouldn't it have shown up in the NDVI?

We could not detect a significant long-term increase in NDVI.

I. 236-238. This is probably the most important statement in the Discussion & Conclusions. This needs to be emphasized

We now better emphasize the importance of this statement, and also refer to it at the end of the abstract.